# Incremental Connected Component Detection for Graph Streams on GPU

Kyoungsoo Bok [1], Namyoung Kim [2], Dojin Choi [3], Jongtae Lim [2] and Jaesoo Yoo [2,*]

1   Department of Artificial Intelligence Convergence, Wonkwang University, Iksandae 460,
    Iksan 54538, Jeonbuk, Republic of Korea; ksbok@wku.ac.kr
2   Department of Information and Communication Engineering, Chungbuk National University, Chungdae-ro 1,
    Seowon-gu, Cheongju 28644, Chungbuk, Republic of Korea; la6803@chungbuk.ac.kr (N.K.);
    jtlim@chungbuk.ac.kr (J.L.)
3   Department of Computer Engineering, Changwon National University, Changwondaehak-ro 20, Uichang-gu,
    Changwon 51140, Gyeongsangnam, Republic of Korea; dojinchoi@changwon.ac.kr
*   Correspondence: yjs@chungbuk.ac.kr; Tel.: +82-43-261-3230

**Abstract:** Studies on the real-time detection of connected components in graph streams have been carried out. The existing connected component detection method cannot process connected components incrementally, and the performance deteriorates due to frequent data transmission when GPU is used. In this paper, we propose a new incremental processing method to solve the problems found in the existing methods for detecting connected components on GPUs. The proposed method minimizes the amount of data to be sent to the GPU by determining the subgraph affected by the graph stream update and by detecting the part to be recalculated. We consider the number of vertices to quickly determine the connected components of a graph stream on the GPU. An asynchronous execution method is used to shorten the transfer time between the CPU and the GPU according to real-time graph stream changes. In order to show that the proposed method provides fast incremental connected component detection on the GPU, we evaluated its performance using various datasets.

**Keywords:** connected component; GPU; graph stream; incremental processing

## 1. Introduction

A graph is a data structure that represents multiple relationships through vertices and edges [1,2]. With the advancements in big data and artificial intelligence technology, graphs are widely used to process and analyze various relationships between objects [3,4]. Recently, with the advancements in real-time application services and convergence services, graph streams have been created in which the vertices and edges that comprise the graph change continuously [5,6]. These graph streams are used in fields such as social media analysis, anomaly detection, fraud detection, and content recommendation [7,8]. Social media platforms such as Facebook, Twitter, and Instagram use graph streams to model human relationships and content usage histories, analyze real-time change characteristics, and recommend information.

Many studies have developed methods for quickly and accurately analyzing graph streams, in which numerous changes occur in real time [9–11]. Connected component (CC) detection is one of the most basic algorithms used in graph analysis [12–16]. A CC must have paths that can connect all vertices to each other in an undirected graph and must not have paths connected to vertices belonging to other CCs. A method that can detect CCs efficiently in a large graph is needed for various applications that use CCs [17,18]. For example, when computing graph clustering, a CC is used as a sub-algorithm, and CC algorithms are frequently called to perform computations. As the size of graphs used in the internet of things, on the web, and in social media has increased, methods for processing large-scale graphs have been developed [19–22]. Additionally, a distributed processing method has been proposed for detecting CCs in large graphs [23–26].

Applications have a characteristic whereby only some of the vertices and edges that constitute the graph change [3,4]. Therefore, methods have been proposed to detect CCs incrementally using partially changed vertices and edges in the graph stream [27]. In general, the incremental method can detect CCs faster than the static method, but it has a limitation when processing large graphs. Graph processing techniques using a graphics processing unit (GPU) have been proposed to perform fast parallel processing on large-scale graph data [17,28–30]. The GPU employs a single instruction, multiple threads (SIMT) architecture to support a high level of parallel processing using multiple cores [31–36]. However, it must perform synchronization when the processing of different thread blocks has finished, and it has limited memory. If the method of processing graph streams using a central processing unit (CPU) is applied as is to a GPU, problems such as memory access and load imbalance will occur. Therefore, an algorithm for the parallel processing of a graph on a GPU with simple computing devices and little memory is needed.

Various methods have been proposed to detect CCs in a GPU for graph streams. GPU-based packed-memory array (GPMA) [30] is a technique used for maintaining the dynamic graph structure in the memory of a GPU according to the packed memory array (PMA) designed for the CPU [37]. Evolving Graph (EvoGraph) [29] employs static and incremental processing methods selectively to compute CCs. GPU-specific ConnectIt (GConn) [17] uses various static and incremental algorithms that can be applied to the GPU for CC computation. GPMA requires more memory than typical arrays and takes more time to build as the size of the update batch increases. In particular, GPMA uses a lock-based approach and has performance degradation when the processing is performed on a single GPU, owing to thread collisions that occur when a large update batch is processed. EvoGraph can detect CCs by employing user functions, but no technique has been proposed for the efficient computation of CCs in the GPU structure. GConn manages disjoint sets based on the union-find data structure to support various types of incremental processing of CCs. However, GConn assumes that all processing is possible on the GPU, whose limited memory characteristics are not considered.

When a large graph is processed on a GPU, a large amount of data transfer between the CPU and the GPU is required. As the GPU is mounted on a Peripheral Component Interconnect Express (PCI Express) socket, it is not physically connected to the CPU or the main memory. To process a graph on a GPU, it is first necessary to copy data from the CPU to the GPU memory to perform the computation and then copy the computation results back to the CPU. Data exchange between the CPU and GPU is performed through PCI Express, and the number of data transfers must be minimized because the data transfer time is longer than the data processing time. However, to detect CCs for a dynamic graph that changes in real time, data must be transferred to the GPU for each processing batch, which causes the problem of frequent data transfers. This paper proposes a method of GPU-based efficient incremental CC detection in a graph stream. The proposed method applies the size to the component labels using a weighted quick-union algorithm to compute CCs quickly and reduce the amount of data transferred to the GPU. We propose a method of transferring the minimal information needed to recalculate the CC to the GPU. Furthermore, the performance degradation caused by the data transfer time is minimized by sending data from the CPU to the GPU while the GPU performs computations through the asynchronous execution method.

The remainder of this paper is organized as follows: Section 2 describes the previously proposed methods; Section 3 explains the proposed method for processing graph streams efficiently on a GPU; Section 4 presents the results of various performance evaluations; and Section 5 concludes the paper.

## 2. Related Work

The simplest method for detecting CCs in an undirected graph is to search each vertex sequentially via the depth-first search (DFS) or breadth-first search (BFS) algorithms [38–40]. The time complexity of DFS and BFS is $O(n + m)$, where $n = |V|$ and $m = |E|$, because

each vertex must be visited and each edge must be traversed. The union-find algorithm was proposed to improve the DFS and BFS methods. It uses a disjoint set structure that stores components partitioned into subsets that do not overlap with each other [41,42]. Each vertex initializes the disjoint set that contains only itself and visits the edge to form the union of the sets, to which two vertices—the endpoints of the edge—belong. If the data structure of the disjoint sets is implemented as an array, the time complexity of union-find is $O(n)$, and if it is implemented as a tree, union-find is faster than $O(n)$. Weighted quick-union is a technique that always connects a smaller tree to a larger tree when performing a union to reduce the time taken to search for the root in union-find [43,44]. In quick-union, when connecting two trees with each node connected to an edge, a large tree may be connected to a small tree. If a larger tree is connected to a smaller tree, the average time for which each node of the entire tree explores the root node increases because the time taken for a node of the larger tree to explore the root node of the smaller tree increases. Weighted quick-union detects CCs by always connecting a smaller tree to a larger tree. Therefore, it reduces the average time taken to reach the root node of the tree from each node.

GPMA is a GPU-based dynamic graph storage method for high-speed analytic processing on GPUs [30]. GPMA maintains the sorted components partially by leaving gaps where the GPU can quickly apply graph changes at a limited gap ratio according to the PMA structure. The update batch is sorted, and the sorted batch is partitioned again. Each partitioned part belongs to the tree's single node. The update is performed according to the size of each partitioned part in the constructed tree. GPMA allows sorted components to access adjacent components quickly and sends the neighborhood of the continuously changing graph to the graph analytics module. Graph analytics perform the query tracking task while interacting with the active graph structure. Meanwhile, the graph stream buffer module performs batch processing of graph streams coming in from the CPU and periodically sends a batch to the graph update module. The graph update module updates the active graphs stored on the GPU with the received batch. The active graph is saved in the PMA-based structure of GPMA and can access adjacent components quickly.

EvoGraph [29] selectively performs static processing and incremental processing for CC detection. If the graph update affects only a small part of the graph, incremental processing is efficient. Conversely, if the graph update affects most subgraphs, it is more efficient to recalculate the entire graph. Therefore, EvoGraph selects a processing method by determining whether static processing or incremental processing is more efficient when an update occurs in the graph. Traditional GraphIn [27] is applied to the GPU to build an inconsistency list by detecting vertices that are affected by the update when an update occurs. Whenever an update batch affects vertices in the graph, the affected vertices are added to the inconsistency list, and the computation is performed again. In the case of a CC, when an update occurs in which an edge is inserted, the endpoints of the inserted edge are checked, and if they are in another CC, the pertinent vertices are added to the inconsistency list. The inconsistency list is built to incrementally compute only the parts affected by the update, and Property Guard is applied to determine whether static or incremental processing is advantageous according to the input graph or the graph algorithm.

GConn [17] was developed for the GPU version of ConnectIt [12], which was proposed to compute the CCs of a graph on multi-core CPUs, and it manages disjoint sets using the union-find data structure. Its purpose is to implement various static and incremental computation algorithms for CCs to determine the effect of algorithm selection on performance and identify the best-performing CC. In incremental CC detection, when a new edge is inserted into a CC, it is processed through the incremental connection algorithm after static GConn processing. The static GConn first initializes each vertex's label with its own ID through InitLabel. The static CCs are then computed through the sampling phase and finish phase presented in ConnectIt. The root's label is assigned to each vertex through finalization to complete the static computation. If an edge is inserted into the CC, the update is processed by performing the finish phase for the incremental processing of the inserted edge. In the finish phase, several variant methods of the union-find algorithm

are used to implement the optimal algorithm, whereby the final component labels are determined. Union-find performs tracking for each disjoint set to allow the CC of each set to have the same label, and it supports three operations: make set, union, and find.

Incremental Gather-Apply-Scatter (iGAS) was proposed, which uses a cost model in the CPU environment to selectively perform static and incremental processing for graph streams [3]. It analyzes the graph processing history to predict the detection and processing costs of the recalculation part and then performs incremental processing if this is advantageous over static processing. PowerGraph's GAS model was transformed to enable incremental processing in the graph stream. The caching technique is used to save the subgraphs that have been read once and the computation results, while the adjacent vertices are pre-patched to increase the incremental processing efficiency.

## 3. GPU-Based Incremental Connected Component Detection

### 3.1. Overview

Methods for determining connected components in graph streams have been used to analyze the relationship between various information generated by the internet of things and social network services in real time. When the graph stream is processed only by the CPU, it is difficult to determine the connected components in real time due to the amount of computation. Recently, methods using GPUs have been proposed for processing connected components in large graph streams [17,29,30]. GPMA consumes more memory than typical arrays, and as the size of update batches increases, performance degrades due to the time it takes to rebuild PMA and GPMA. In particular, GPMA as a lock-based approach causes performance degradation when processing on a single GPU due to the collision of threads during large update batch processing. EvoGraph builds an Inconsistency List to gradually calculate only the parts affected by the update and determines based on the input graph or graph algorithm whether gradual processing is beneficial or static processing is beneficial by applying the Property Guard. However, EvoGraph puts a heavy burden on developers by allowing users to specify user functions to compute connected components, and no techniques have been proposed to compute efficient connected components in the structure of GPUs. GConn has proposed a variety of static and incremental algorithms that can be applied on GPUs for efficient connected component computation. However, GConn does not take into account the low-capacity memory size of the GPU and requires an optimized technique to minimize memory latency on the GPU.

The existing CC detection methods have a performance degradation problem, either because the CC is not processed incrementally or because data is exchanged frequently between the GPU and the CPU. When a GPU is used to detect a CC for a graph stream, the following two cases should be considered. The first is the data exchange between the GPU and the CPU. The data must be copied from the main memory and delivered to the GPU to perform computational operations on the GPU. Changes occur in the graph stream in real time, and data must be copied to the GPU frequently to compute the updates. When a large amount of data is transferred, the memory capacity of the GPU is exceeded, and data are copied more frequently. Second is the case of the core idle state, in which the GPU or the CPU is in an idle state without performing operations. The CPU does not perform operations while the CPU transfers data to the GPU, but the GPU does perform operations. When the GPU transfers data to the CPU, neither the CPU nor the GPU performs operations. The GPU and the CPU are different devices, and they can achieve more efficient processing if the asynchronous execution structure is used. In this paper, we propose a novel incremental CC computation technique to solve the problem of processing graph streams using the GPU. To reduce GPU computational costs and CPU-GPU communication costs, we determine the regions affecting existing CC results in the input graph stream and gradually determine CC. In addition, it uses a weighted quick union method to quickly calculate CC. The overall processing performance is improved by reducing the idle time of the CPU through the asynchronous processing method.

In this paper, we deal with and undirected graph streams $G_t = (V_t, E_t)$ without weight for detecting CCs, where $t$ represents the input time of $G$, $V_t$ represents the set of vertices at time $t$, and $E_t$ represents the set of edges at time $t$. A graph stream may have both vertices and edges added and deleted. Insertion and deletion of isolated vertices that are not connected to other vertices do not affect the determination of connected components. It is the insertion and deletion of edges that affect the actual connected components in the graph stream. If an edge is deleted, it is easy to determine whether the CC is changed by calculating whether a path exists with neighboring vertices within the CC where the starting and ending vertices of the deleted edge are the same. However, when an edge is inserted, the vertices constituting the CC may be combined with other CCs in addition to increasing. As a result, it is the insertion of edges that requires a lot of computational cost to determine the progressive connected components of CCs in the graph stream. Therefore, we describe the processing process of the proposed method, focusing on the insertion of edges rather than the deletion of edges.

Figure 1 shows the processing architecture of the proposed incremental CC detection method performed on the GPU. The proposed method incrementally updates the CC results by identifying the edges that affect the previous CC results when a new graph stream is input and an existing generated CC result exists. When the first graph stream is input, the incremental processing cannot be performed because the initial CC result does not exist. The dual path execution determines whether a CC result calculated through a graph stream exists and determines whether static or gradual processing exists. If no CC result exists, the initialization phase generates the first CC result using the existing static method. The first CC result is assigned a component label, and then stored on the CPU as a connected component-labeled graph (CCLG). As the graph changes over time for the graph stream, the region of the graph that requires recalculation is determined according to the change. In the computation for an update, the entire graph is not computed again each time the graph changes, and the part that is affected by the update is determined through the recalculation part identification (RPID). RPID compares the update batch with the CCLG to determine the region where recalculation is required, generating a recalculation list (RL), which is then sent to the Graph Update module of the GPU. The Graph Update module performs the CC update computation for the RL and constructs and sends Result_L—the computation result—to the CPU. The transferred Result_L is merged with the existing saved CCLG to finish the update process.

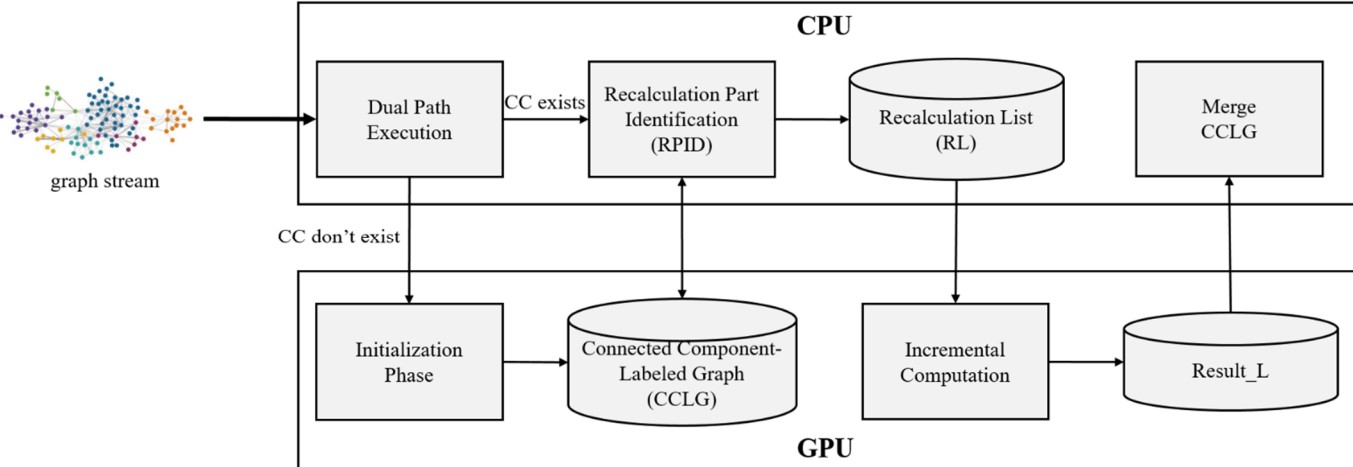

**Figure 1.** Overall processing architecture of the proposed method.

A typical synchronization-based GPU processing method involves transferring all the graphs that need to be processed from the CPU to the GPU, which uses multicore to work in parallel, terminating all the operations it is currently working on, and then initiating a new task. Therefore, the synchronization technique results in latency when

the CPU does nothing while the GPU performs operations, and the workload allocated to the GPU is unbalanced. In addition, the synchronization technique generates frequent communication between the GPU and the CPU to transmit the results of the performance performed by the GPU and the subgraph to be performed next. In particular, sending all the edge lists to the GPU to recalculate CC in the graph stream increases the amount of data transmission and increases the amount of unnecessary computation in the GPU. To solve these problems, asynchronous structural processing is required to reduce unnecessary data transmission and allow GPUs and CPUs to perform tasks simultaneously. The proposed method first identifies the edge affecting the existing CC results when a graph stream is input to reduce the communication cost to be transmitted between the CPU and the GPU and unnecessary computation on the GPU, and then transmits only the related RL to the GPU. In the synchronization technique, the CPU analyzes the newly entered graph stream and generates RL in advance while the GPU performs the operation to solve the problem that the CPU cannot perform the operation until the GPU finishes the operation.

Figure 2 shows an example of asynchronous execution of the proposed method. The proposed method checks for stored graphs in the first double-path run when a graph stream is inserted. If CCLG does not exist, the initial CCLG is constructed by computing the first CC by sending the entire input graph to the GPU, since the proposed method is to process the first input graph. In addition, if we have previously stored CCLGs, we configure them to determine the areas affected by the update due to the graph stream inserted through the RPID module and to send RLs to the GPU. The RL is transmitted to the GPU to perform incremental processing. When a new graph stream is input after transmitting RL from the CPU to the GPU, the CPU asynchronously performs RPID before receiving the updated CC result to configure the new RL. The GPU updates CC via the Graph Update module according to the RLs received, and the CPU sends the new RLs to the GPU. When the CC update is completed in the GPU, Result_L is sent to the CPU. The CPU merges with the existing CCLG, and at the same time, the GPU updates CC through the new RL frame and transmits the Result_L to the CPU.

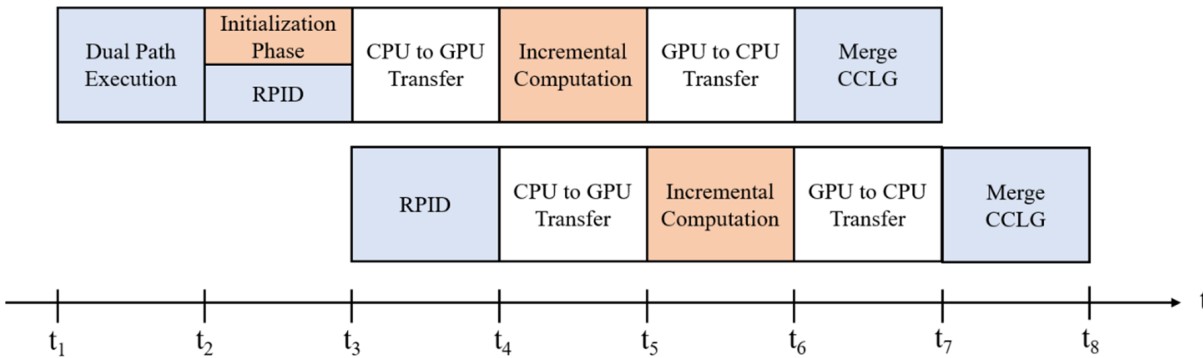

**Figure 2.** Asynchronous execution.

### 3.2. Component Label Management

The CC results calculated in the initialization phase are managed by the CCLG table to determine the recalculation area according to the graph stream input. When the CCs are first computed, the graph stored in the edge list structure is changed and stored in the form of the vertices' unique numbers and component labels. Figure 3 shows the CCLG table containing the initially computed CC results, where ID is the vertex identifier included in the CC and the component label is the identifier of the CC. As shown in Figure 3a, when CCs exist, Figure 3b is a table that manages information about vertices, and Figure 3c is a CCLG table. In Figure 3c, the component label is an identifier for the CC containing each vertex, which means that vertices with the same component label have paths that connect to each other through edges and have no connecting paths with vertices belonging to other CCs. The initial component label is set to the label of the calculated CC of the first

input graph stream. If the connectivity of the previous graph is changed by the insertion of the graph stream, the CCLG table is checked to determine where the CCs must be computed again.

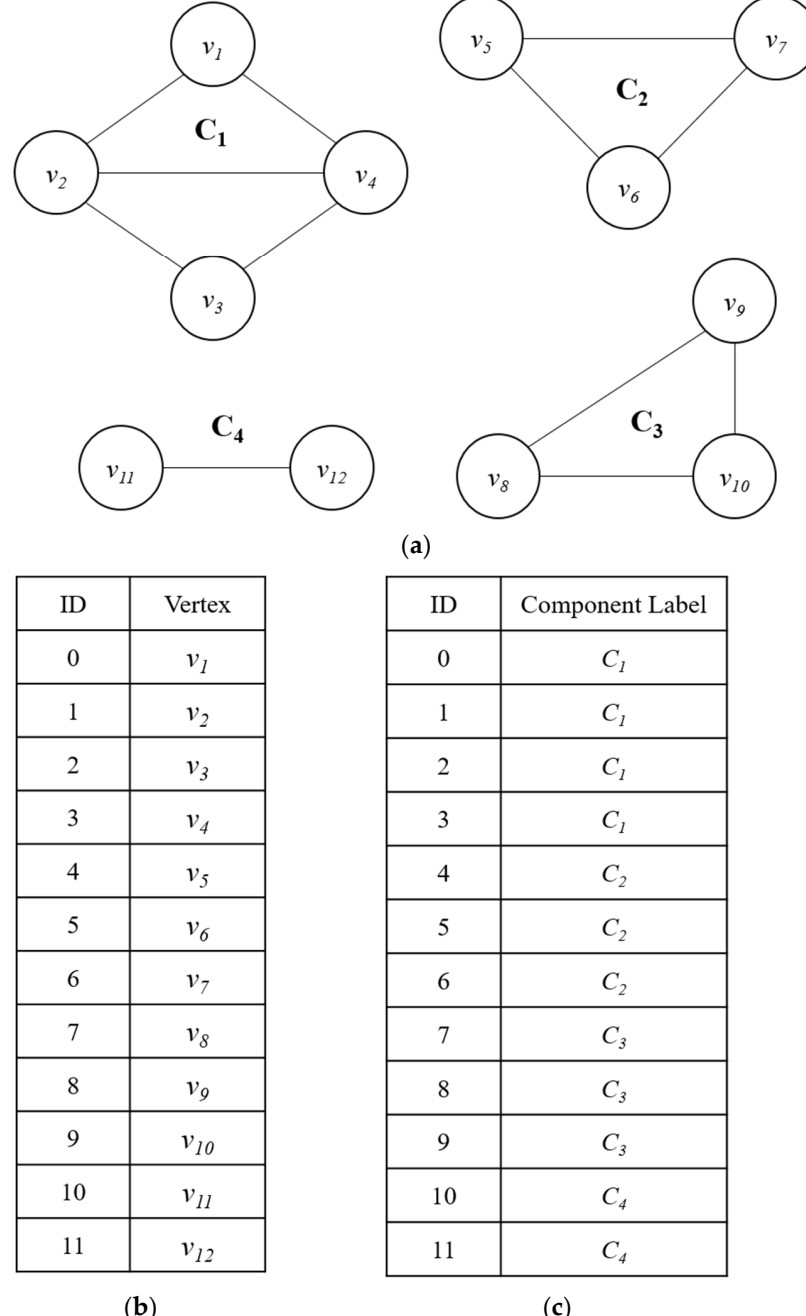

(a)

| ID | Vertex |
|----|--------|
| 0 | $v_1$ |
| 1 | $v_2$ |
| 2 | $v_3$ |
| 3 | $v_4$ |
| 4 | $v_5$ |
| 5 | $v_6$ |
| 6 | $v_7$ |
| 7 | $v_8$ |
| 8 | $v_9$ |
| 9 | $v_{10}$ |
| 10 | $v_{11}$ |
| 11 | $v_{12}$ |

(b)

| ID | Component Label |
|----|-----------------|
| 0 | $C_1$ |
| 1 | $C_1$ |
| 2 | $C_1$ |
| 3 | $C_1$ |
| 4 | $C_2$ |
| 5 | $C_2$ |
| 6 | $C_2$ |
| 7 | $C_3$ |
| 8 | $C_3$ |
| 9 | $C_3$ |
| 10 | $C_4$ |
| 11 | $C_4$ |

(c)

**Figure 3.** Example of the initial results of CCs. (**a**) Input graph stream. (**b**) Vertex table. (**c**) CCLG table.

If the graph structure changes due to the input of the graph stream, the component label of the existing CCLG table is changed to quickly determine the recalculation area of the CC. When a graph change such as the insertion of an edge occurs, the part that affects the CC is determined, and the CC is recalculated. Here, a representative vertex is selected to send only the information required for computation to the GPU. When there are $n$ CCs in graph G, the CC set is denoted as $C = \{C_i | i \in n\}$. Each CC are represented as $C_i = \{E_{i1}, E_{i2}, \ldots, E_{im}\}$, where $m$ is the number of vertices belonging to each CC. The representative vertex $R_i$ of each $C_i$ is the first element of $C_i$ is the root vertex of $C_i$, which is defined as $E_{i1} \in C_i$ and $E_{i1} = R_i$. To distinguish a particular vertex from a regular vertex if it

is a representative vertex of CC, the proposed method sets the number of vertices contained in CC in the component label as negative. If a particular vertex is not a representative vertex of CC, it sets the ID of the representative vertex in the component label.

Figure 4 shows the CCLG table changed to the component label storage form after the initialization phase. In the CCLG table, the representative vertex of each CC is updated with a negative number, and the remaining vertices have the ID of the representative vertex. For example, when there are four CCs in the entire graph, the root vertices of the CCs are $v_1$, $v_5$, $v_8$, and $v_{11}$. The representative vertex of the CC containing $v_1$, $v_2$, $v_3$, and $v_4$ is $v_1$. The component label $v_1$ has a value of $-4$, as the number of vertices contained in the CC is 4. The component labels of the non-root vertices $v_2$, $v_3$, and $v_4$, with the exception of $v_1$, have a value of 0.

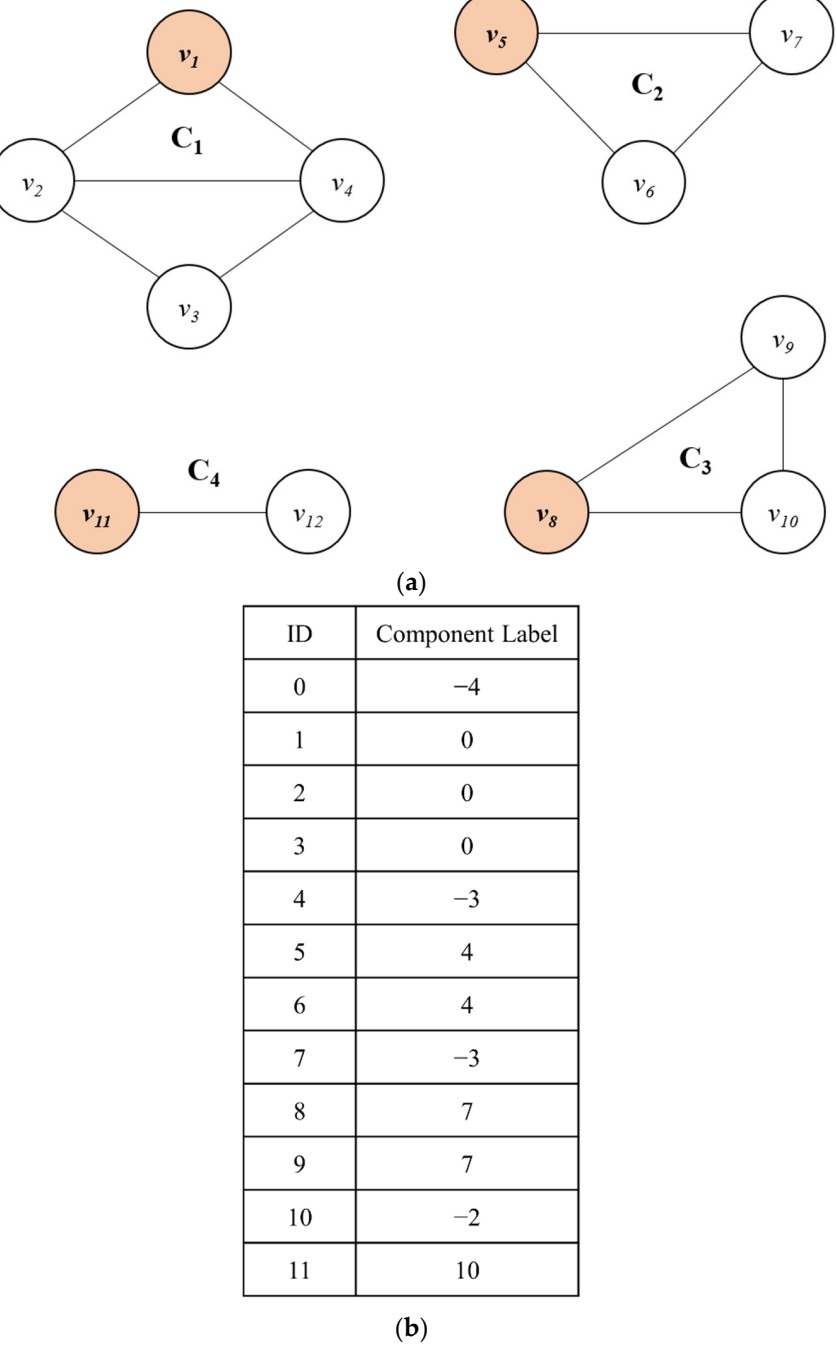

(a)

| ID | Component Label |
|----|-----------------|
| 0 | −4 |
| 1 | 0 |
| 2 | 0 |
| 3 | 0 |
| 4 | −3 |
| 5 | 4 |
| 6 | 4 |
| 7 | −3 |
| 8 | 7 |
| 9 | 7 |
| 10 | −2 |
| 11 | 10 |

(b)

**Figure 4.** Example of CCLG table management. (**a**) Graph example. (**b**) CCLG table.

### 3.3. Incremental Computation

For the determination of the update part, the part that needs to be computed is ascertained and sent to the GPU when a graph stream is entered. In the case of static computation, when an update occurs, the CCs are computed again for the entire graph. If the entire graph is recalculated, a large amount of data must be managed in the GPU memory. Therefore, if a large graph is entered, the memory capacity of the GPU will be exceeded. In this case, the graph is transferred multiple times, which is inefficient because the transfer time is long. Therefore, we need a process for determining the part affected by the update and then constructing an RL to be sent to the GPU. The update of a graph stream is defined as an edge list insertion $(v_s, v_d)$, where $v_s$ is the starting vertex of the inserted edge, and $v_d$ is the destination vertex of the inserted edge. When an edge is inserted, the parts for which recalculation will be performed are determined. Here, the determination criterion is that recalculation must be performed when the CCs of the $v_s$ and $v_d$ of each inserted edge are different. If the vertices $v_s$ and $v_d$ of the inserted edge belong to the same CC, the CC does not change, implying that there is no need to perform a recalculation. However, if the $v_s$ and $v_d$ of the inserted edge belong to different CCs, a path between the different CCs is generated, changing the CCs. If the CCs are changed, a recalculation is required, because the component labels must be updated.

Figure 5 shows an RL generated using RPID, which determines the part affected by the update when an updated graph stream is input. First, when a new edge list is inserted, the component labels of $v_s$ and $v_d$ are checked in the previously constructed CCLG. If the component labels differ, they are added to the RL. The purpose of RPID is to minimize the amount of data to be transferred to the GPU for CC recalculation. A root-based RL is built to minimize the amount of data to be transferred to the GPU. The proposed data-transfer minimization method finds the root label of the component label in the CCLG if the component label in the RL is not a representative vertex and then changes the label value in the RL to the value of the root label. If the vertex of the CCLG is not the representative vertex, the root vertex of the CC can be found quickly because the stored label of the pertinent vertex is the representative vertex's ID. The root's ID is also added to the RL so that the CC can be recalculated on the GPU and the updated CC can be applied quickly on the CPU. Figure 5a shows an example of edges $(v_4, v_6)$ and $(v_8, v_{12})$ inserted into the graph of Figure 4. If two edges are inserted, the component labels of $v_s$ and $v_d$ in each edge list are found in the CCLG. The $v_4$ and $v_6$ labels are $v_{4,L} = 0$ and $v_{6,L} = 4$, respectively. The component labels of an edge $(v_8, v_{12})$ are determined in the same way. Figure 5b shows an example of building an RL. If the component label found is not a negative number, it is not the root label. Therefore, the vertices that have the same IDs as the component labels 0 and 4 of $v_4$ and $v_6$ in the CCLG are found $(v_1, v_5)$, and the label values of the CC are replaced with the labels $(-4, -3)$ of the pertinent vertices to build a final RL to be sent to the GPU. The constructed RL is the table on the right in Figure 5b. The RL constructed through the proposed RPID sends only the root ID and the component labels to the GPU, providing minimal information for computing the CC on the GPU. By transferring minimal data, a large update can be performed efficiently in parallel processing with the limited memory of the GPU.

When the RL is transferred to the GPU, a Graph Update is performed, which efficiently changes the component labels. The absolute value is compared via weighted quick-union for the component label of the root vertex in the RL constructed to calculate the CC efficiently. The larger component label is defined as the large component (LC), and the smaller component label is defined as the small component (SC). Figure 6 shows the process of changing component labels on the GPU. As the component labels of the root ID 0 and the root ID 4 are $-4$ and $-3$, respectively, the root ID 0 is the LC, and the root ID 4 is the SC. To change the component labels of $LC_0 = -4$ and $SC_4 = -3$, the value of $SC_4$ is added to the value of $LC_0$, and the value of $LC_0$ becomes $-7$. After the value of $SC_4$ is transferred again to the CPU, it is changed to 0—the ID value of $LC_0$—to update the values of the CCs of $SC_4$ in the CCLG to $SC_4 = 0$. The root IDs 7 and 10 are computed in the same way.

As the component labels of root IDs 7 and 10 are $-3$ and $-2$, respectively, the results of comparing the sizes are $LC_7 = -3$ and $SC_{10} = -2$. The computation results of adding the smaller component label to the larger component label are $LC_7 = -5$ and $SC_{10} = -7$. The result of changing the values of the LC and SC is reflected in Result_L, i.e., the result table of the graph update process. After Result_L is constructed, it is sent back to the CPU for merging with the original graph.

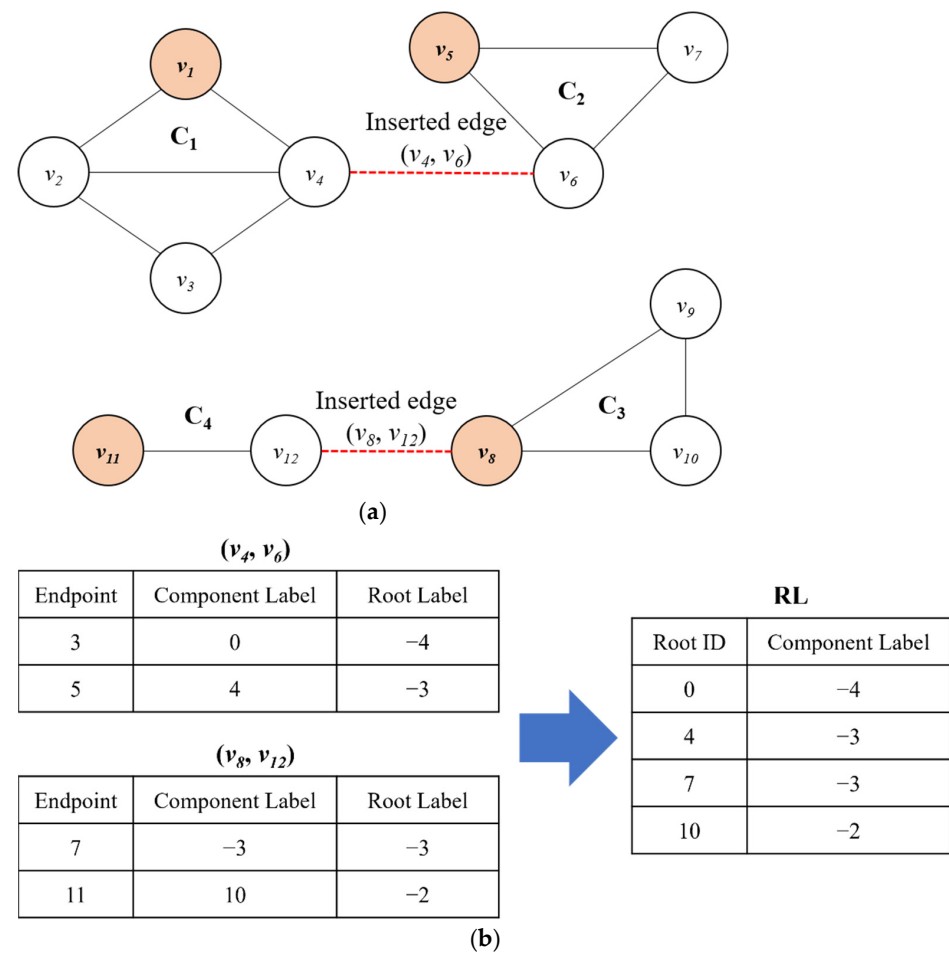

**Figure 5.** Example of the RL generating process. (**a**) Example of edge insertion. (**b**) RL.

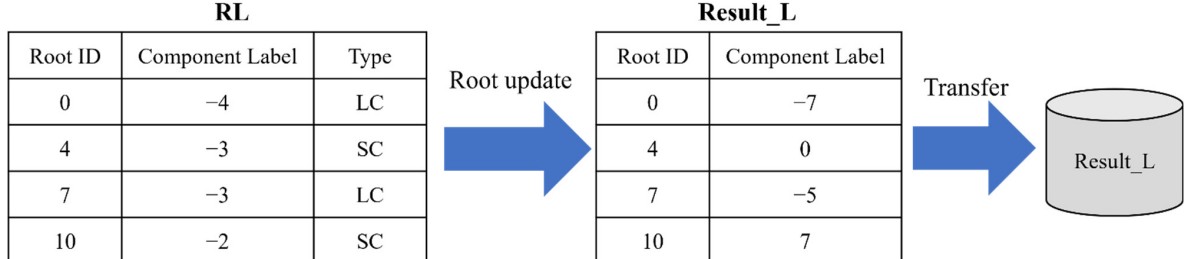

**Figure 6.** Example of a CC update.

### 3.4. Result Merging

As graph streams are continuously input, new RLs are constructed, and the result of updating the RL should be reflected in the labels of the CCLG to indicate the part to be changed. Result_L—the computation result in the Graph Update module—is merged with the original graph and sent to the CPU to perform the merging process with the CCLG stored on the CPU. The component labels to be changed can be accessed quickly through

the root ID in Result_L. To merge the result, an ID in the CCLG that matches the root ID of Result_L is found, and the component label of the pertinent ID is changed to that of Result_L. In the case where the component label of the existing CCLG is changed from a negative number to a positive number, the representative vertex points to another CC, indicating that the component labels of all vertices contained in the CC to which the root vertex belongs are changed.

Figure 7 shows the merging process of Result_L and the CCLG, where Result_L is the CC update result of the two inserted edges $\{(v_4, v_6), (v_{12}, v_8)\}$. Figure 7b shows an example of the result of merging the graphs. First, an ID in the CCLG that matches the root ID of $v_4$ in Result_L (i.e., 0) is found. The component label of $v_1$ with an ID of 0 in the CCLG is updated from $-4$ to $-7$. Similarly, the component label with an ID of 4 in the CCLG is updated from $-3$ to 0. As the component label has updated from a negative number to a positive number, ID 4 is no longer a representative vertex, and the component labels of all the vertices of the CC where ID 4 was the representative vertex have become 0. $v_5$, $v_6$, and $v_7$ belong to the CC where $v_1$ is the root vertex. Similarly, the merge is performed for IDs 7 and 10. After the merge, $v_5$ is contained in the CC of $v_1$, and $v_{11}$ is contained in the CC of $v_8$, indicating that the CCs have been changed owing to the update.

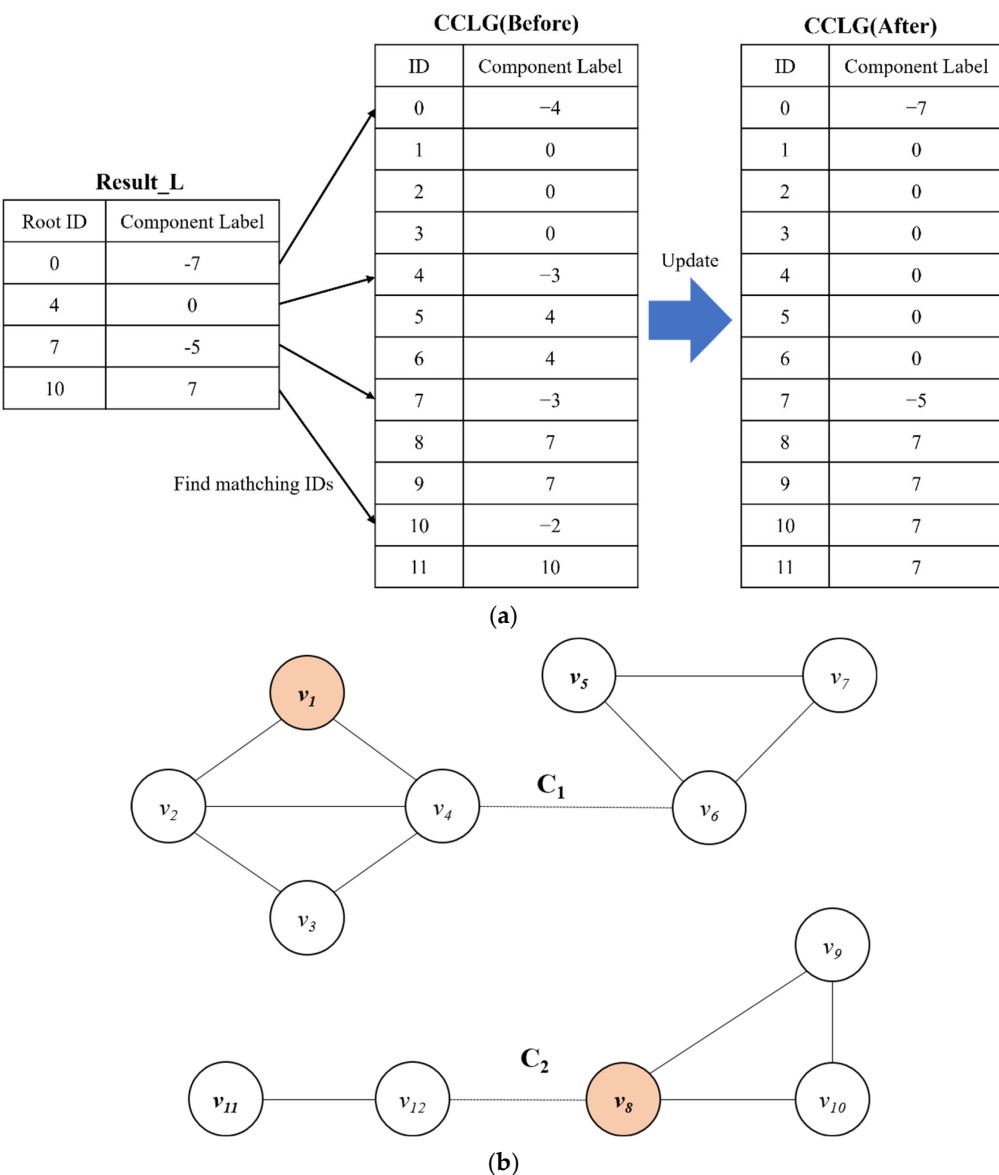

**Figure 7.** Example of merging CCs. (**a**) Merge process. (**b**) Merge result.

## 4. Performance Evaluation

The superiority of the proposed method was demonstrated by evaluating its performance in comparison with the existing methods. We implemented performance evaluation using the Python 3.8.12 (CPU) and CUDA 11.2 (GPU) languages on an Intel(R) Core(TM) i7-9700 CPU @ 3.6 GHz and a GeForce RTX 3060 device. The proposed method determines the incrementally connected components in the graph stream. For experimental evaluation, we performed it using two real graph datasets, soc-Pokec and soc-Live Journal1, which have dynamic change characteristics in the form of graph streams among the actual graph datasets provided in SNAP [45]. Table 1 shows the characteristics of the dataset used in performance evaluation. The experimental data comprised a dataset of 1.6 M vertices and a dataset of 30.6 M edges. The random dataset was used to test the performance in an environment with many CCs. In the case of the random dataset, a portion of the dataset was used for the initial graph, and random sampling was performed in batches. In the performance evaluation, the data in the form of an edge list was inserted as a stream. At certain time intervals, the update batch was inserted by changing the batch size between 50 and 13,000,000, and the size of the inserted batch was set according to the purpose of the performance evaluation. Furthermore, tests were performed in which the percentage of the update batch was varied to measure the performance change according to the percentage of the graph stream. The part that performed the initialization phase of CCs, which is the preprocessing step of the proposed incremental process, was implemented using cuGraph—a RAPIDS application programming interface (API).

**Table 1.** Dataset.

| Datasets | # of Vertices | # of Edges |
| --- | --- | --- |
| Random | 1.60 M | 30.6 M |
| soc-Pokec | 1.60 M | 85.70 M |
| soc-LiveJournal1 | 4.85 M | 30.6 M |

Figure 8 shows the ratio of the CC processing time, data transfer time, and GPU kernel launch time for the datasets. We measured the average processing time for 5000 updates for an edge list with an update batch size of 50. We measured the percentages of the processing time, kernel launch time, and data transfer time by setting the total processing time to 100%. The data transfer time was the average of the sum of the data exchange times between the CPU and the GPU. The processing time was the sum of the time for building and merging the RL on the CPU and the time for computing CCs for the update batch on the GPU. GPUs perform the same kind of operations in parallel across multiple cores based on the single instruction, multiple thread (SIMT) model. When a specific task is performed on the CPU, the requested task is divided into several partitions, and the GPU kernel is launched. GPU kernel launch is the total latency to run the kernel, which is the initialization overhead required to start the kernel to perform the graph stream on the GPU. In experimental evaluation, GPU kernel launch is measured only once, when a graph stream request is first delivered. The random dataset, soc-LiveJournal1 dataset, and soc-Pokec dataset accounted for 37.78%, 33.82%, and 17.20% of the processing time and 5.08%, 5.33%, and 4.00% of the data transfer time, respectively. The GPU kernel launch times were 56.13%, 60.85%, and 78.8% for the random dataset, soc-LiveJournel1 dataset, and soc-Pokec dataset, respectively. GPU kernel launch time accounts for a large proportion of small dataset processing but a relatively small proportion of large graph stream processing since it is performed only once when processing initial graph streams. The performance evaluation results indicated that the CC computation time varied depending on the dataset.

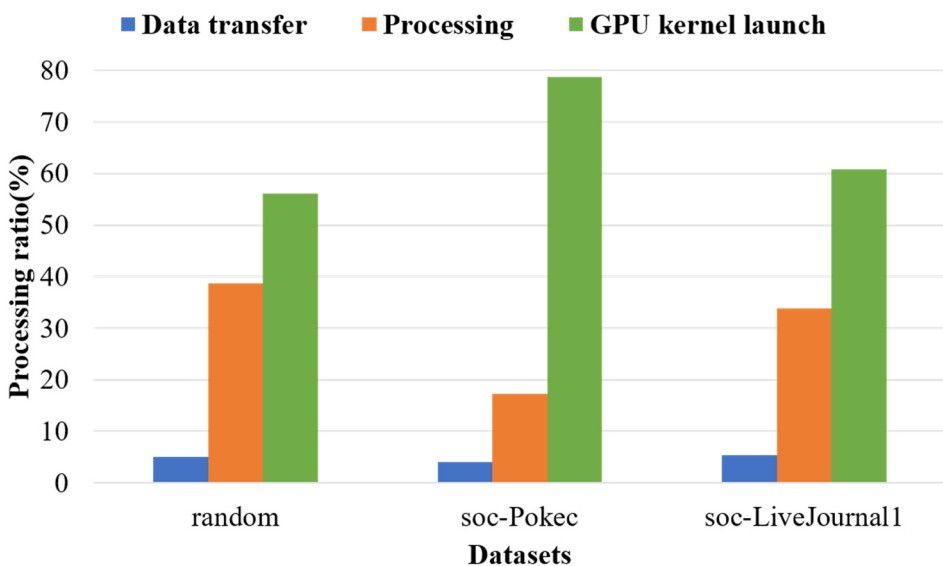

**Figure 8.** Processing ratio for datasets.

The processing time of the proposed method on the GPU and the data transfer time between the CPU and the GPU were evaluated. Figure 9 shows the incremental CC computation time on the GPU, the GPU kernel launch time, and the data sending and receiving time between the CPU and the GPU. The random dataset was used, the update batch size was set to 50, and the experiment was performed with inputs of 5000 update batches in 5 s intervals. GPU processing refers to the time taken for the update module to process incremental CCs on the GPU. Data transfer refers to the time required to transfer RL from the CPU to the GPU and the time required to transfer Result_L from the GPU to the CPU. Kernel refers to the time that the kernel was first started on the GPU. GPU Kernel execution time was mostly constant regardless of the size of the data, and when processing large graph streams, kernel execution time was only processed once, so it did not account for a large percentage. However, when processing small graph streams, GPU kernel execution time used most of the total processing time.

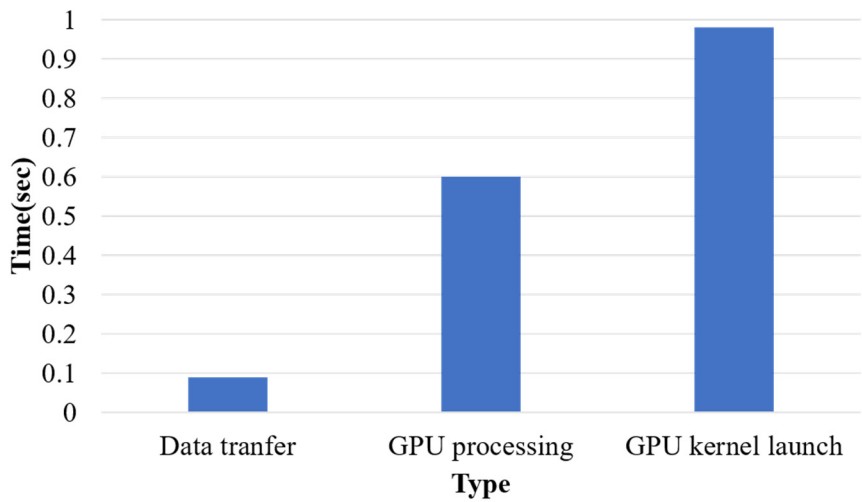

**Figure 9.** Data transfer and GPU processing time comparison.

In the case of computing CCs in a graph stream environment, there is static and incremental computation. The processing time was compared between the existing static computation method and the incremental computation method according to the size of the update batch. The random dataset was used, of which 50% was used for initial graphs

to generate graph streams for CC computation, while the remaining 50% was inserted by setting the percentages according to the criteria. Among the static processing methods, a CC detection method using a sequential, parallel approach was implemented on the CPU, while the GPU method used the RAPIDS cuGraph API to compute the CC. Figure 10 shows the results of comparing the total processing time between static computation and incremental computation. The performance evaluation time of the proposed method includes the preprocessing time on the CPU. In the case of static computation, the processing time increased as the batch size increased because the CCs had to be computed again every time an update occurred on the GPU. In the case of the proposed method, the preprocessing time for finding the representative vertices to reduce the amount of data to be sent from the CPU to the GPU and constructing the proposed structure was included in the processing time. Therefore, the proposed method took slightly longer than the existing static computation method the first time a graph stream was input, but even when the proportion of the update batch size increased, there was almost no impact on the processing time. The total processing time of the proposed method was reduced by 3086% and 358% compared with that when static computations were performed on the CPU and the GPU, respectively. The performance evaluation results confirmed that the size of the update batch affected the performance. The total processing time was improved by 92%, 461%, 445%, and 445% when the batch size was 20%, 50%, 70%, and 90%, respectively. This confirms that incremental processing outperforms static processing as the batch size increases. Therefore, even if the batch size is large, the proposed incremental processing method facilitates efficient CC detection.

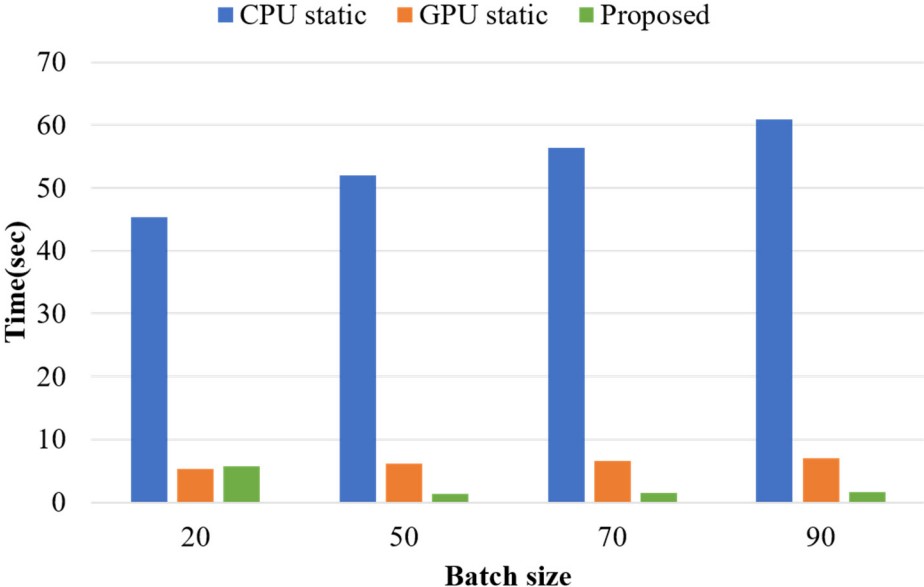

**Figure 10.** Total processing time according to batch size.

Figure 11 shows the results of comparing the performance of the proposed method with the existing static processing method on the GPU by measuring only the update processing time on the GPU for the batch sizes shown in Figure 10. For the existing GPU-based static processing method, we measured only the time required to process the update on the GPU, which did not include the preprocessing time for merging the update batch into the original graph. The proposed method exhibited an average improvement of 229% compared with the existing GPU-based static processing method. The improvements were 206%, 230%, 237%, and 243% when the batch sizes were 20%, 50%, 70%, and 90%, respectively. As the static method processes a graph by adding an updated batch to the original graph, the processing time is proportional to the batch size. Since the proposed method performs the update by determining only the recalculation part, there is little change in the processing time, even if the size of the inserted batch increases.

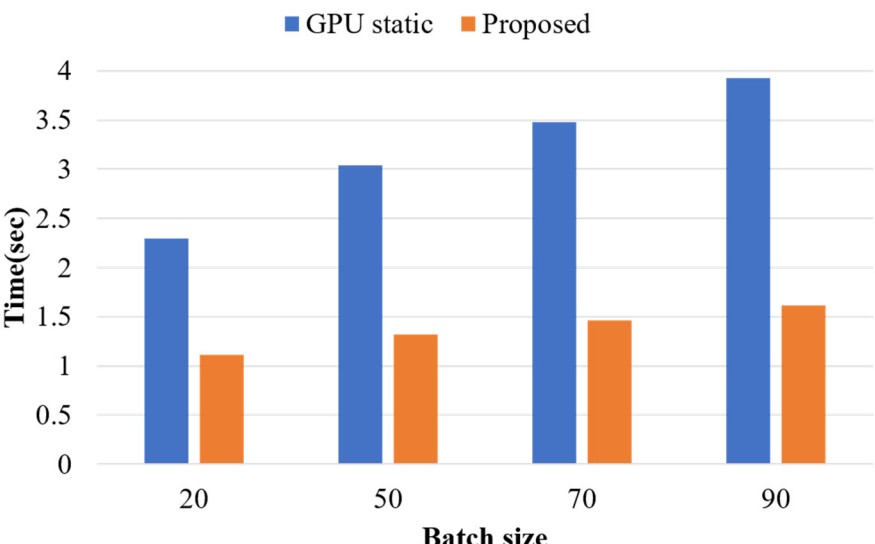

**Figure 11.** Incremental processing time per batch.

The graph stream processing time was measured for the incremental CC computation to compare its performance with that of the existing method. Figure 12 shows the processing times of the existing and proposed methods for incremental CCs where an update batch of 50 is inserted. The processing times of the existing EvoGraph method [29] and the proposed method were compared for an update batch of a certain size. The incremental processing time per batch exhibited an average improvement of 337% compared with the static computation time. The criterion for selecting static or incremental processing was as follows: if the sum of two root component labels in the RL was $\geq 1.06$ times the largest root component label in the entire graph, static processing was performed; otherwise (i.e., <1.06 times), incremental processing was performed. For 10,000 edge sets of the random dataset, 5000 update batches were inserted. The proposed method exhibited an average processing time improvement of 128%.

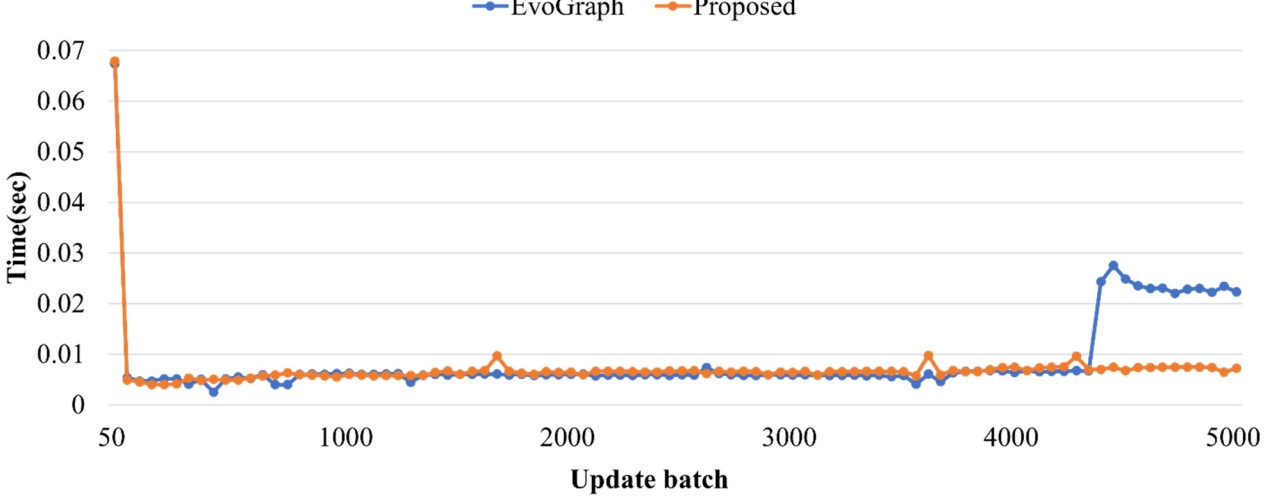

**Figure 12.** Processing time for incremental CCs with a batch size of 50.

## 5. Conclusions

In this paper, we propose an efficient incremental CC detection method in graph streams using a GPU. In the proposed method, the size is applied to component labels to manage CCs efficiently. When a graph stream is input, the RPID module identifies the part affected by the graph and generates an RL for the part that requires CC recalculation.

An RL is built, and asynchronous data transfer is performed to reduce the data transfer time between the GPU and the CPU. In a performance evaluation, it was proved that the performance of the proposed method is superior to that of the method of recalculating the entire graph on the GPU and the CPU, respectively. In the future, we will conduct performance comparative analysis with techniques that perform various approaches on various experimental datasets and conduct research to improve the progressive processing performance for CC in multiple GPU environments. In addition, we will conduct a study to apply the proposed method to directional graphs and a study on graph partitioning techniques to improve GPU parallel processing performance.

**Author Contributions:** Conceptualization, K.B., N.K., D.C., J.L. and J.Y.; methodology, K.B., N.K., D.C. and J.Y.; software, N.K. and D.C.; validation, N.K., D.C. and J.L.; formal analysis, K.B., N.K., D.C. and J.Y.; investigation, K.B., N.K., D.C. and J.L.; resources, N.K., D.C. and J.L.; data curation, N.K. and D.C.; writing—original draft preparation, K.B. and N.K.; writing—review and editing, K.B., N.K. and J.Y.; visualization, N.K. All authors have read and agreed to the published version of the manuscript.

**Funding:** This work was supported by the National Research Foundation of Korea (NRF) grant funded by the Korea government (MSIT). (No. 2022R1A2B5B02002456); an Institute of Information & Communications Technology Planning & Evaluation (IITP) grant funded by the Korea government (MSIT) (No.2014-3-00123, Development of High Performance Visual BigData Discovery Platform for Large-Scale Realtime Data Analysis and No. 2021-0-02082, CDM_Cloud: Multi-Cloud Data Protection and Management Platform); and the "Cooperative Research Program for Agriculture Science and Technology Development (Project No. PJ016247012022)" Rural Development Administration, Republic of Korea.

**Institutional Review Board Statement:** Not applicable.

**Informed Consent Statement:** Not applicable.

**Data Availability Statement:** Not applicable.

**Conflicts of Interest:** The authors declare no conflict of interest.

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
