# Peer review of "Incremental Connected Component Detection for Graph Streams on GPU"

_electronics, doi:10.3390/electronics12061465_

Round 1

Reviewer 1 Report

This manuscript presents a new incremental processing method for connected component detection on GPU. The method basically seems reasonable to the reviewer and, according to the evaluation, it outperforms an existing incremental processing method EvoGraph in some cases.

The reviewer thinks the most serious shortcoming of this manuscript is that the effect of execution pipelining (Fig. 2) is not properly discussed and/or evaluated. Pipelining does not always work well: it generally requires each processing stage to take almost the same amount of time. If not, a stage with the largest processing time becomes a performance bottleneck and the latency of each task becomes long. Since such an inbalance of processing times was observed (according to Figs 8 and 9), an increase of the latency may have occurred. Comparison with the case where the pipelining is not applied is highly recommended. If we don't have to worry about it in theory, a proper explanation should be added.

In addition, the definition of the CC components in Section 3.2 is difficult to understand, as the variable $i$ is already used as an index of the CC set. You can make it clearer by redefining the CC components in the $i$-th CC set as, for example, $C_i = \{e_{i,j} | 1 \leq j \leq m_i\}$, where $m_i$ is the number of the components in the $i$-th CC set. The subsequent definitions should also be revised.

Author Response

Dear Reviewer,

We would like to sincerely thank you for your attentive indications and good comments. Our paper is partially rewritten in order to revise and complement your comments. please refer to the attached file. 

Many thanks.

Jaesoo Yoo

Reviewer 2 Report

Incremental Connected Component Detection for Graph 2 Streams on GPU 

This paper proposes a new approach to detect connected components (CC) in streaming graphs. Rather than recomputing the CC everytime the graph changes, this approach only requires processing a sub-graph. The approach also makes use of both CPU and GPU. 

- The approach is not very novel. 

- The writing needs a lot of improvement. It takes a lot of patience to understand the current paper. 

- The experimental section is underwhelming. 

-I believe this paper has merit. Although it is not currently publishable, I think the authors can improve the writing and experimental section to make it more suitable for publication. 

Questions/comments to Authors: 

  1. 1. One of my biggest complaints is that the writing is not pedagogical and is loosely written. Below, I am pointing out some of the issues that I found. The entire paper needs to be revisited to add clarity. Moving examples to the beginning, say in the overview section, can greatly help. 

  1. 2. Line 195-197: Does your approach work with directed graphs? Does it support edge deletion? What about vertex deletion? Explicitly mention what is supported or not supported. 

  1. 3. Change the overall architecture to an overview. Explain things at a high level, without going into details; examples will help. 

  1. 4. Line 200: At this point, the reader has no idea what “dual path” execution is. 

  1. 5. “Static computation module”: The usage of the word “static” does not seem appropriate. Maybe something like “initialization phase” is more meaningful. 

  1. 6. Line 214-224: These lines are talking about an optimization, which appears in the middle of a section that explains the approach. It would be better to explain the approach first (say GPU only) and then talk about optimizations. 

  1. 7. Line 249: “Component labels are the CC label” → CC labels are not defined at this point. 

  1. 8. Fig 3: What is the importance of ID in table 3? Why not just use vertex_id? If ID is vertex_id, why save both? 

  1. 9. The first element of the set “Ci” is called “ei” and is the representative element of a set. Mathematically, a set is not ordered, and hence “first” does not mean anything. I understand what the authors meant, but the definition needs to be improved. Moreover, the first use of the representative element comes in line 261 but is defined much later. 

  1. 10. Line 262: “The representative vertex is marked” → How is it marked? Which data structure is updated? 

  1. 11. As of now, the component label can be (i) The label of CC; (ii) the number of vertices in CC, or (iii) the ID of the representative vertex. This is very confusing. Maybe use “CC ID” in table 3 and #vertices/representative_id in table 4. 

  1. 12. The experimental evaluation section is very weak. 

  1. 13. Why were only two real graphs chosen? Show results for all graphs benchmarked in other works such as GUNROCK. Also add at least one more platform. 

  1. 14. “The GPU kernel launch” → This was very confusing to me. To be fair, the authors do explain what this means in the paper. However, “kernel launch overhead” is a widely used term in HPC that refers to the overhead of just launching an empty kernel. In this work, this term has a completely different meaning. Something like “initialization overhead” is more appropriate. 

  1. 15. [minor] Line 195: we deals → we deal 

Author Response

(The authors gave the same response as above.)

Round 2

Reviewer 1 Report

The reviewer confirmed that the manuscript was properly revised by incorporating the comments on the previous version. The reviewer understood the authors' viewpoint that the proposed execution was more like "asynchronous" than "pipelined." The manuscript now became acceptable for publication.